# Sports Event Image, Satisfaction, Motivation, Stadium Atmosphere, Environment, and Perception: A Study on the Biggest Multi-Sport Event in Indonesia during the Pandemic

Kenius Kogoya [1], Tri Setyo Guntoro [2,*] and Miftah Fariz Prima Putra [2,*]

1   Postgraduate School of Social Sciences, University of Cenderawasih, Jayapura 99351, Indonesia; kogoyakenius2@gmail.com
2   Department of Sport Sciences, Faculty of Sport Sciences, University of Cenderawasih, Jayapura 99351, Indonesia
*   Correspondence: trisguntoro09@gmail.com (T.S.G.); mifpputra@gmail.com (M.F.P.P.)

**Abstract:** The National Sports Week (known in Bahasa Indonesia as *Pekan Olahraga Nasional* or PON) consumes a large budget as Indonesia's largest multi-sports event. This raises the question of whether it is only a sporting event or has an impact on society. Studies related to multi-sports events, specifically in the form of local or small-scale, such as PON, are still limited in Asia. The aim of this study was to investigate six important domains (constructs) of the 20th PON (PON XX) held in Papua in 2021: sports event image, motivation, satisfaction, stadium atmosphere, environment, and perception of the impacts. In addition, this study assessed the correlations between those six constructs and associations between the explanatory variables (gender, distance of residence, and involvement in the sporting event) and those six constructs. A pre-tested questionnaire was used to assess those six constructs and the explanatory variables. We included 675 respondents aged between 17–57 years, with a mean age of 22.87 years in the study. The results showed that the implementation of PON XX was positively received by the community and their highest motivation to watch this event was for entertainment. The involvement of the participant in the event was significantly associated with sports event image, satisfaction, motivation, stadium atmosphere, environment, and the perception of PON impact. The distance of the residence from the venues was only significantly associated with the perceived impact of PON XX on the community while gender had no association with all six constructs. There was a strong correlation between the other five investigated constructs and the perceived impact of PON XX.

**Keywords:** sports event; National Sports Week (PON); sports event image; motivation; satisfaction; stadium atmosphere; stadium environment; perception of the impact

## 1. Introduction

The development of sports in Indonesia has a long historical record up to the modern era after independence, precisely in 1947. Although the Indonesia Olympic Committee was formed in 1947, Indonesia was unable to participate in the 1948 London Olympics (Putra 2021). This is because of a protracted course to satisfy management necessities wherein the visas for Indonesian athletes and officials needed to be issued through the Dutch government for which they were not very helpful (Lutan 2005). This eventually became an important moment for the development of sports in the country. Sri Sultan Hamengubowono, chairman of the Indonesia Olympic Committee at the time, initiated the holding of a National Sports Week or known in Bahasa Indonesia as *Pekan Olahraga Nasional* (PON) on 9 September 1948, in Solo (Lutan 2005). Presently, this event is still held regularly by competing in various sports and it later became the largest multi-sport event in the country (Putra and Ita 2019). The 20th PON (PON XX) which occurred on October 2021 in Papua province was a new event for Indonesian sports because apart from being

held during the coronavirus disease 2019 (COVID-19) pandemic, it was also held in four different clusters (cities/regencies). Therefore, the Vice President of Indonesia at the closing ceremony said that the event was the most difficult multi-event sports event in the history of the country's sports journey.

PON is often used as a barometer to measure the development and progress of sports nationally (Guntoro and Putra 2021; Wandik et al. 2021). It involves various competitions and is a miniature of events, such as the Southeast Asian Games, Asian Games, as well as the Olympiads. Furthermore, it is deliberately carried out by the government because aside from having historical value for the Indonesian nation, it is also a sports coaching event that gathers participants and athletes from all provinces (Putra and Ita 2019).

Studies related to sporting events have rapidly increased in recent years. Chen et al. (2021) studied marathon running events but only in one city; Milovanović et al. (2021) explored small-scale sporting and single events, namely the World Championship. Furthermore, Duan and Liu (2021) examined the satisfaction of spectators running a marathon in a small-scale sporting event in Wuhan, China, while Duan et al. (2020) investigated the motivation, satisfaction, and behavioral intention of marathon runners in China. Girish and Lee (2019) examined brand experience, sports event image, and loyalty in ultramarathons; Wann et al. (2008) analyzed the motivation of supporters in different sports, while Kim and Chalip (2004) studied the FIFA World Cup event, specifically regarding the motives, background, interests, and constraints. Moreover, Kim et al. (2017) investigated the economic impact of formula one events, Briedenhann (2011) identified the expectations in the economy and the tourism sector from the community, while Preuss (2005) determined the economic impact of various sporting events. Rozmiarek et al. (2021) also investigated the motivation of European Games volunteers, Wilson (2006) analyzed the economic impact of swimming events, while Lai (2018) studied the Olympics in Beijing, China, but only the event image and destination image were analyzed. Another study by Waitt (2003) examined the social impact of the Olympics and Konstantaki (2008) investigated the socio-cultural impact of the Olympics from the perspective of lecturers and students. Brown et al. (2016) investigated several aspects related to psychology in the Olympic event, while Kavetsos and Szymanski (2010), as well as Dolan et al. (2016), examined life happiness and community satisfaction in mega-event sports. Furthermore, Konstantaki et al. (2019) revealed public opinion related to the theme and content of the Olympics opening ceremony. Madden and Crowe (2002) also examined the economic impact of the Olympics. Mitchell and Stewart (2015) analyzed the economics of tourism in hosting sporting events, while Lamla et al. (2014) investigated the economic impact of the EURO football event.

In general, studies related to the sports events above can be classified into two categories, namely single and multi-event. The size is further divided into three, namely mega-events, such as Olympic Games, medium-sized, such as a national championship, and small-scale, e.g., local or regional level (Kaplanidou and Vogt 2006). Although the PON XX was a multi-sport event and the largest in the country's history (Guntoro and Putra 2021), it is included in the small-scale category because it is a domestic event. The literature has mostly discussed mega sports events, such as the Olympics and the World Cup, while the smaller events have not been widely investigated (Jeong et al. 2019, 2020). The PON XX is very interesting because the budget spent to organize the sporting event was very large, with an estimated amount of ten trillion rupiahs (Guntoro and Putra 2021). Second, Papua province, where the PON XX was held, has continuous security issues due to the actions of the armed terrorist criminal groups. In addition, the event was conducted amid the COVID-19 pandemic that has disturbed many human aspects in Indonesia (Fahriani et al. 2021).

Although issues related to sports events have been widely examined, there are limitations in previous studies. First, no study has jointly discussed public perceptions regarding the impact of sports events related to internal aspects, such as sports event image, satisfaction, and motivation, as well as external aspects including stadium atmosphere and environment in an in-depth manner. Second, similar to other events that require a large budget (Mitchell and Stewart 2015), PON also consumes a fantastically large budget raising

the question of whether it is only a sports event or has an impact on the community. Third, investigations on local sports or small-scale events are limited to single events. These limitations constitute a knowledge gap. Studies on multi-events carried out in Asia also focus on mega sports events, such as the Southeast Asian and Asian Games, as well as the Olympics, while local and multi-sports events have not been examined. Hence, there is a need to identify the socio-cultural perspective from the east (Asia) for a more comprehensive picture related to sports events. Therefore, the aims of this study were: (1) to describe six important domains (constructs) of the PON XX in terms of the sports event image, community motivation to watch the event, community satisfaction, stadium atmosphere, stadium environment, and the community perception towards the impacts of the event; (2) to assess the correlations between the constructs and (3) to investigate the associations of the explanatory variables (gender, distance of residence, and involvement in the sporting event) and the constructs.

## 2. Materials and Methods

### 2.1. Study Setting and Study Sites

A cross-sectional study was conducted between 2 and 15 October 2021 in Papua, Indonesia where the PON XX took place. Papua is an Indonesian province with 547 islands and the largest area of 312.224 km$^2$ or 16.64% of the total land area in the country (BPS 2022). Based on the Human Development Index (HDI), the province is at the bottom with an average score of 60.63 in the last three years, far below the national average of 72.05 (BPS 2022). Security issues have become a major concern in Papua due to the actions of armed terrorist criminal groups which greatly disturb security and public order.

The PON XX was held in four districts/cities: Jayapura City, Jayapura, Merauke, and Mimika. This study was conducted in all these districts/cities during the COVID-19 pandemic.

### 2.2. Study Participants

The individuals who were around venues in those locations (spectators, committees, community members, athletes or trainers) were approached and asked to participate in this study. The respondents were sampled using a purposive sampling method to include both groups of respondents who were directly involved in the events (such as committees, community members, athletes and trainers) and those who were not involved directly, such as spectators.

This study involved 675 participants aged between 17–57 years with a mean age of 22.87 years (SD ± 5.34). All the participants were citizens living in Papua. The demographic characteristics of participants are presented in Table 1.

**Table 1.** Respondent demographics (n = 675).

| Category | Frequency | Percentage (%) |
|---|---|---|
| Gender | | |
|   Female | 269 | 39.9 |
|   Male | 406 | 60.1 |
| Distance from residence to venues | | |
|   ≤1 km | 208 | 30.8 |
|   2–4 km | 198 | 29.3 |
|   ≥5 km | 269 | 39.9 |
| Involvement in PON XX | | |
|   Involved | 139 | 20.5 |
|   Not involved | 536 | 79.4 |
| Occupation | | |
|   Contract/honorary employee | 124 | 18.3 |
|   Student | 187 | 27.7 |
|   Civil servant | 76 | 11.2 |
|   Indonesian Army/Police | 43 | 6.3 |
|   Entrepreneur | 138 | 20.4 |
|   No answer | 107 | 15.8 |

### 2.3. Study Instrument, Constructs and Variables

A pre-tested instrument was used to assess the six constructs of the study: image, satisfaction, motivation to watch, stadium atmosphere, atmosphere environment, and the perception of the impacts. All these constructs were classified as response variables in this study.

To assess the image related to the implementation of PON, the Sports Event Image (SEI) instrument developed previously by Kaplanidou and Vogt (2007) was adapted and modified. SEI initially had 13 items in the form of a semantic differential scale with a range of alternative answers from 1 to 7. Three years later, Kaplanidou and Vogt (2010) retested the SEI and released two items in the initial version. In this study, the initial version of 13 items was used because the other two items are still relevant in the context of PON XX in Papua. The test on 68 communities found that one item (Healthy Unhealthy) had a low correlation coefficient value with $r = 0.202$ and $p < 0.30$. However, considering that the PON XX was held during the COVID-19 pandemic, it was still used. For other items, the validity and reliability values ranged from 0.341 to 0.637 and 0.675 to 0.710, respectively, indicating that the items were acceptable to be used.

Community satisfaction while watching PON XX was measured by the Sports Audience Satisfaction Scale (SASS), modified from the instrument developed by Huang et al. (2015), Lita and Ma'ruf (2015), and Škorić et al. (2021). A total of eleven items were selected by considering the high factor loading value and suitability with the context of the event. The test found a range of validity and reliability values between 0.634–0.775 and between 0.914–0.921, respectively. Each question of SASS has five alternative answers in the form of a Likert scale ranging from completely unsatisfied (1) to completely satisfied (5).

The next four constructs were assessed using the instruments consisting of the questions with five alternative answers ranging from strongly disagree (1) to strongly agree (5). To determine the community's motivation for watching the event, the instrument developed by Snelgrove et al. (2008) and Balaji and Chakraborti (2015) was used and modified using three subscales, namely entertainment, aesthetics, and vicarious achievement. These factors were translated into eleven items. The validity values ranged between 0.677–0.831 and reliability values ranged between 0.944–0.949.

To measure the perception of the impact of the PON XX on the community, a Questionnaire on the Impact of Sports Events on the Community (QISEC) developed by Guntoro and Putra (2021) was used. QISEC consists of 26 items. In this study, simplification and re-testing were carried out on ten items with high validity values, namely four items in economic factors, as well as three each in psycho-social and infrastructure factors. The validity and reliability values ranged from 0.694 to 0.838 and 0.933 to 0.940, respectively.

To assess the stadium atmosphere, a modification was carried out on the Stadium Atmosphere Scale (SAC) (Balaji and Chakraborti 2015) which consists of different aspects, such as physical layout, facilities aesthetics, entertainment experience, and social interaction. Our study used four items with a high factor loading value only, which were tested in the field. The test found validity and reliability values of 0.704–0.799 and 0.832–0.868, respectively.

The Stadium Environment Scale (SEC) (Cho et al. 2019) was used to assess the stadium environment. It consists of five subscales: parking, cleanliness, fan control, food service, crowding, and desire to stay. In this study, only six items with a high factor loading value were taken. The validity and reliability values ranged from 0.317–0.749 and 0.759–0.861, respectively.

In addition, some demographic variables (gender, involvement in the event, distance of the residence to the venues and occupation) were collected. Involvement in the event indicated whether the respondents were involved directly during the PON XX as a committee, security, athletes and trainer or as a spectator only. In this study, we considered gender, involvement in the event and distance from the residence to the venues as potential explanatory variables, therefore, their associations with all six constructs were assessed.

*2.4. Data Collection Procedures*

The initial stage was to adapt the language and modify the instrument according to the objectives. The language adaptation stage involved English and Indonesian language experts who were independent and unrelated to this study. Subsequently, the Indonesian version of the instrument was completed, and all instruments were tested on 68 communities in Papua. The initial test was conducted online using a Google form where the link was distributed to the public. The results of this validity and reliability test were used to modify the final study instrument section.

After the study instrument was declared valid and reliable, 15 final-year sports students were recruited as enumerators. After the intensive training on data collection procedures in the field, they were deployed in the areas where the PON XX event was held. Each prospective respondent was given an explanation related to the purpose and description of this study and then asked for their approval. The respondents answered each of the questions and they could ask the question to the enumerators if they had difficulty understanding the question. All respondents provided their written informed consent prior to participating in the study.

*2.5. Statistical Analysis*

The mean, standard deviation, and percentage of the data were provided descriptively. To assess the relationship between variables, Pearson's product-moment correlation analysis was used. All data analyses were performed using the IBM SPSS version 26 program (IBM Corp, Armonk, NY, USA).

## 3. Results

*3.1. Descriptive of the Assessed Constructs (Response Variables)*

The presentation of results started with descriptive, followed by relationship analysis between constructs, and associations between explanatory variables, such as gender, the distance of the residence from the venues and the involvement of the respondent in the PON XX and the six constructs. The descriptive and normality analysis of all six constructs assessed in this study are presented in Table 2. Our data suggested that the data of all six constructs had a normal distribution.

**Table 2.** Descriptive analysis and normality of the data.

| Variable | Min | Max | Mean | SD | Skewness | Kurtosis |
|---|---|---|---|---|---|---|
| PON image | 13 | 91 | 55.24 | 18.91 | −0.735 | −0.433 |
| Community satisfaction | 11 | 55 | 39.38 | 13.71 | −0.547 | −1.031 |
| Community motivation | 11 | 55 | 39.86 | 13.94 | −0.654 | −0.992 |
| Stadium environment | 6 | 30 | 20.47 | 7.13 | −0.355 | −1.004 |
| Stadium atmosphere | 4 | 20 | 14.63 | 5.10 | −0.598 | −0.980 |
| PON impact perception | 10 | 50 | 35.20 | 11.74 | −0.585 | −0.875 |

For the image construct, the results presented in Table 3 show that the community described the PON on the positive side, with attributes, such as fulfilling (42%), excellent (42%), stimulating (36%), joyful (43%), healthy (38%), exciting (46%), valuable (43%), beautiful (50%), adventurous (39%), relaxing (39%), inspiring (46%), cheerful (45%), and supportive (46%).

In terms of community satisfaction with the implementation of the PON, the results suggested that more than half of respondents (58.20%) stated that it was satisfactory, while 26.83% had different opinions (Figure 1A). Concerning people's motivation to watch the PON XX, 53.18% was related to entertainment, 27.92% to aesthetics, and 18.90% to vicarious achievement (Figure 1B). Concerning the perception of the PON XX impacts on the economic, psycho-social, and infrastructure aspects, the community responded that the economics had a greater impact than the other two aspects (Figure 1C).

**Table 3.** The community images of PON XX.

| Domain | Semantic Scale | | | | | | | Domain |
|---|---|---|---|---|---|---|---|---|
| | 1 | 2 | 3 | 4 | 5 | 6 | 7 | |
| Unfulfilling | 18 | 11 | 4.6 | 5.8 | 11 | 8.9 | 42 | Fulfilling |
| Poor | 20 | 6.2 | 4.7 | 5.2 | 10 | 12 | 42 | Excellent |
| Stimulating | 36 | 18 | 10 | 6.7 | 8.1 | 6.1 | 16 | Unstimulating |
| Sad | 18 | 9.2 | 5.9 | 5 | 8.9 | 9.6 | 43 | Joyful |
| Healthy | 38 | 19 | 9 | 9.3 | 7.1 | 5.3 | 12 | Unhealthy |
| Boring | 14 | 9.6 | 4.4 | 4.7 | 9.8 | 11 | 46 | Exciting |
| Valuable | 43 | 23 | 7.1 | 5.9 | 5.5 | 4.3 | 12 | Worthless |
| Ugly | 13 | 8.3 | 6.7 | 5.9 | 5.9 | 10 | 50 | Beautiful |
| Unadventurous | 21 | 9.6 | 4.4 | 8 | 9.3 | 8.7 | 39 | Adventurous |
| Distressing | 13 | 8.1 | 8.4 | 11 | 11 | 9 | 39 | Relaxing |
| Inspiring | 46 | 19 | 8.6 | 6.7 | 4.4 | 3.9 | 11 | Uninspiring |
| Gloomy | 13 | 10 | 5.6 | 6.2 | 8.7 | 11 | 45 | Cheerful |
| Unsupportive | 16 | 8.1 | 5.3 | 6.7 | 7.6 | 10 | 46 | Supportive |

All values are in percentage (%).

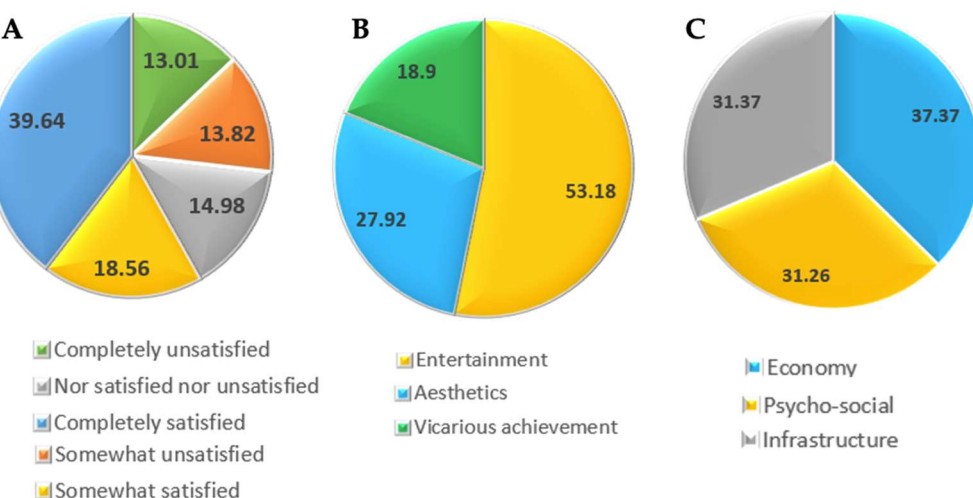

**Figure 1.** Community satisfaction in the implementation of PON XX (**A**), motivation to watch PON XX (**B**) and perception of the impact of PON XX in Papua (**C**).

The stadium environment constructs, including aspects of parking, cleanliness, and food, were judged to be on the positive side (Table 4). Consequently, the aspect of "happy at the venues" had a high percentage of agree (19.70%) and strongly agree (38.22%). Concerning the size of the venues, the responses were relatively balanced.

The stadium atmosphere was rated positively (Table 4). Concerning the "signs that are quite adequate", 48.29% of respondents answered agree and strongly agree. Meanwhile, for "artistic decoration", 51.26% answered agree and strongly agree. For "viewing in the venues as a pleasant experience", 52.6% answered agree and strongly agree, while on the "sociable" aspect, 55.11% tend to agree and strongly agree.

**Table 4.** Detailed responses of stadium environment and stadium atmosphere constructs.

| Statement | Strongly Disagree | Disagree | Neutral | Agree | Strongly Agree |
|---|---|---|---|---|---|
| **Stadium environment** | | | | | |
| The venues have a large parking lot | 12.00 | 14.07 | 16.00 | 19.70 | 38.22 |
| The area around the PON XX venues is kept clean | 12.59 | 13.93 | 16.74 | 18.96 | 37.78 |
| Spectators/supporters who behaved rudely/annoyingly were monitored by the security system around the venues | 13.48 | 12.44 | 17.48 | 20.30 | 36.30 |
| The food and drinks offered around the venues are delicious | 10.67 | 16.89 | 22.37 | 18.52 | 31.56 |
| The venue is too small | 23.26 | 21.04 | 20.15 | 12.89 | 22.67 |
| I feel happy to linger in these venues | 12.00 | 14.07 | 16.00 | 19.70 | 38.22 |
| **Stadium atmosphere** | | | | | |
| Signs (entrance/exit, toilet, parking, and soon) inside the venues are adequate | 14.96 | 19.56 | 19.56 | 33.33 | 14.96 |
| The decorations inside the venues look artistic | 14.52 | 17.19 | 20.44 | 36.74 | 14.52 |
| Watching the match/contest on venues is a fun experience | 13.19 | 15.41 | 21.48 | 39.41 | 13.19 |
| I enjoy being able to socialize with other fans/spectators during PON matches/contests | 11.26 | 14.81 | 19.11 | 43.85 | 11.26 |

### 3.2. Correlations between Response Variables

A summary of the correlations between response variables is presented in Table 5. All variables had a significant correlation with the perception of PON impacts. Satisfaction with the stadium environment had the strongest correlation (r = 0.901, *p* < 0.01).

**Table 5.** Correlation test results between variables.

| | | 1 | 2 | 3 | 4 | 5 | 6 |
|---|---|---|---|---|---|---|---|
| 1. | Perception of the impact of PON XX | 1 | 0.812 ** | 0.820 ** | 0.799 ** | 0.812 ** | 0.822 ** |
| 2. | Community satisfaction | | 1 | 0.897 ** | 0.820 ** | 0.901 ** | 0.906 ** |
| 3. | Community motivation to watch | | | 1 | 0.820 ** | 0.857 ** | 0.865 ** |
| 4. | Image of PON XX | | | | 1 | 0.789 ** | 0.810 ** |
| 5. | Stadium environment | | | | | 1 | 0.901 ** |
| 6. | Stadium atmosphere | | | | | | 1 |

** Significant at the 0.01 level (2-tailed).

### 3.3. Factors Associated with Response Variables

The associations between explanatory variables (gender, involvement of the respondent in the PON XX and the distance from residence to the venues) and response variables (six constructs) are shown in Table 6. Our data suggested that gender had no association with six response variables. In contrast, there were significant associations between the involvement in the PON with all six constructs of response. The distance from residences and the PON venues was associated with the perception of the impact of the PON XX only (Table 6).

**Table 6.** Factors associated with response variables.

| Variable | Perception of Impact | | Sport Image Event | | Satisfaction | | Motivation | | Stadium Environment | | Stadium Atmosphere | |
|---|---|---|---|---|---|---|---|---|---|---|---|---|
| | Mean ± SD | F | Mean ± SD | F | Mean ± SD | F | Mean ± SD | F | Mean ± SD | F | Mean ± SD | F |
| Gender | | 0.624 | | 0.225 | | 0.067 | | 0.647 | | 0.574 | | 0.250 |
| Male | 34.91 ± 11.87 | | 54.96 ± 19.23 | | 39.49 ± 13.87 | | 40.21 ± 14.23 | | 20.64 ± 7.30 | | 14.04–15.05 | |
| Female | 35.64 ± 11.57 | | 55.96 ± 18.44 | | 39.21 ± 13.48 | | 39.33 ± 13.51 | | 20.22 ± 6.87 | | 14.15–15.35 | |
| Involvement | | 26.909 ** | | 26.186 ** | | 31.150 ** | | 37.148 ** | | 18.089 ** | | 28.661 ** |
| Yes | 39.71 ± 4.23 | | 63.63 ± 4.44 | | 45.04 ± 3.34 | | 46.12 ± 9.46 | | 22.73 ± 4.23 | | 16.04–17.26 | |
| No | 34.03 ± 5.22 | | 53.07 ± 6.69 | | 37.91 ± 5.27 | | 38.24 ± 14.46 | | 19.88 ± 5.43 | | 13.65–14.55 | |
| Distance of residence | | 3.693 * | | 2.936 | | 2.083 | | 2.481 | | 1.588 | | 2.803 |
| ≤1 km | 33.63 ± 12.44 | | 52.63 ± 20.73 | | 37.92 ± 14.44 | | 38.14 ± 14.50 | | 19.75 ± 7.69 | | 13.24–14.73 | |
| 2–4 km | 36.79 ± 11.01 | | 56.14 ± 18.08 | | 40.67 ± 12.42 | | 41.10 ± 12.97 | | 20.91 ± 6.31 | | 14.50–15.80 | |
| ≥5 km | 35.23 ± 11.57 | | 56.61 ± 17.88 | | 39.55 ± 13.97 | | 40.27 ± 14.12 | | 20.71 ± 7.23 | | 14.13–15.35 | |

* Significant at the 0.05 level. ** Significant at the 0.001 level.

## 4. Discussion

This study investigated six constructs of the biggest sporting event in Indonesia in 2021 (PON XX): community image, satisfaction, motivation to watch, stadium atmosphere, atmosphere environment, and the perception of the impacts. This study also investigated the associations between gender, involvement in the event, and distance from the residence to the venues using those six constructs.

### 4.1. PON XX Constructs

The participants who witnessed PON XX tended to give a positive image. Concerning the "ugly-beautiful" dimension, the majority of participants rated the PON XX as beautiful. This is not surprising because a variety of world-class sports infrastructures have been built in the province. In addition, the natural beauty of the province is also considered to have contributed to this context. According to Widiastono and Angriani (2018), the area has several tourist attractions, including natural, cultural, historical, and sports tourism. The "inspiring and exciting" dimension had the same value and this indicates that the community is very inspired by the PON XX. This is because the event, which was previously considered very difficult to organize due to geographical conditions, human resources, supporting infrastructure, security, and the COVID-19 pandemic, turned out to be successful. Consequently, the efforts made by the Provincial Government, the Indonesian National Sports Committee, and the committee have inspired the community. President Joko Widodo praised and appreciated the efforts of the Provincial Government and the committees for the successful implementation of the event. "Papua can" and "people can do it [meaning we all can too]" were the words of the President. This indicates that the PON XX has inspired and attracted various parties due to the limitations and difficulties faced in the process of organizing the event.

Regarding community satisfaction, the majority of respondents were satisfied with the event. This result is in line with another study on larger sporting events (Kavetsos and Szymanski 2010). According to Dolan et al. (2016), sporting events have a positive emotional impact, specifically on the dimensions of happiness and life satisfaction of the spectators. Similarly, Mitchell and Stewart (2015) who analyzed mega sports events found that people who live where sporting events are held will feel happy, satisfied, and proud. In the context of the PON XX, the respondents reported this event has been successfully organized by the government and the committee.

For the motivation of the audience, the results found that entertainment purpose was the main motivation. This is in line with previous studies which examined audience motivation in sporting events. Wiid and Cant (2015) found entertainment motivation to be the main and highest score compared to other dimensions. A similar result was found not only in multi-event sports but also in single events, where the entertainment dimension was one of the main motivations for the audience to watch the match (Wann et al. 2008). Although during the COVID-19 pandemic, the motivation to watch the PON XX did not decline because it was not the biggest national sports entertainment but also Papua was trusted to host the event for the first time (Guntoro and Putra 2021; Wandik et al. 2021).

Concerning the stadium environment, the respondents tended to positively rate the various aspects, such as parking, cleanliness, and food around the venues. A total of 57.92% tended to enjoy lingering around the venues. Cho et al. (2019) stated that the stadium environment has a direct impact on the desire to stay and even return to the place where an event was organized. Cleanliness, large parking lots, and other supporting facilities around the venues are well-conditioned during this event. A similar result was found in the stadium atmosphere, where aspects related to "being able to socialize" had the highest scores. This is because the PON is the largest multi-sport event in the country (Putra 2021; Guntoro and Putra 2021; Putra and Ita 2019) which involves all the best athletes, hence, a very large number of spectators often come to witness the event. A large audience for the

PON match will facilitate more interaction in the community. This is expected to establish togetherness and brotherhood among the community.

On the perception of sporting event impacts on the community, the majority of respondents considered the PON XX to have a greater impact on the economics than the infrastructure and psycho-social aspects. Previous studies on sports events have also confirmed this (Kim et al. 2017; Briedenhann 2011; Kaplanidou and Vogt 2010; Wilson 2006). Sports events will attract a large number of spectators thereby increasing economic benefits for the host community (Preuss 2005). Apart from the economic impact, this study also found that infrastructure was highly perceived by the community, this is because after Papua was designated as the host of the PON in 2014 (Putra and Ita 2019), developments, such as venues, hotels, roads, and support were developed. Therefore, it is not surprising that the infrastructure aspect has the second-highest score because people associate the PON XX with the development carried out by the government. Konstantaki et al. (2019) and Thomson et al. (2013) showed that sports events have potentially broad impacts on society, not only in the economic aspect but also on infrastructure, culture, and others. For the perception of psycho-social impact, the value was not as high compared to the other two aspects. According to Dolan et al. (2016), sporting events will make the people of the region/country feel proud. This is because there is a kind of acknowledgment and trust in the region/country from an external scope.

### 4.2. Correlations between Response Variables

Our analyses found there was a strong relationship between sports event image, motivation, satisfaction, stadium atmosphere, environment, and the perception of impacts. This is supported by Lita and Ma'ruf (2015) and other studies which reported that the public perception of the impacts of sports events, in terms of economy, infrastructure, and psycho-social, is influenced by several factors, such as sports event image (Kaplanidou and Vogt 2007), motivation (Duan et al. 2020), satisfaction (Duan and Liu 2021; Brown et al. 2016), stadium atmosphere (Balaji and Chakraborti 2015), and environment (Cho et al. 2019). Consequently, the community's response to the impact of the PON XX is multi-dimensional. For the aspect of audience satisfaction, the stadium environment and atmosphere variables had the highest coefficient values. This indicates that audience satisfaction is closely related to aspects of the atmosphere around and inside the stadium. What spectators see and feel in and around the stadium will have a significant impact on satisfaction.

### 4.3. Factors Associated with Response Variables

Our data suggested that respondents' involvement in PON XX was significantly associated with all six constructs (responses variables). The average scores among respondents who were involved in the PON XX event were significantly higher in all constructs compared to those of non-involvement respondents. This is because involved respondents received a direct impact from the committees, athletes, officials, and security compared to the group that was not directly involved. The residents who were directly involved tended to actively feel the impacts of the event (Hallmann and Zehrer 2012).

The distance of residence from the venues of PON XX was significantly associated with the perception of the impact of the PON XX suggesting that the distance of residences contributes to shaping public perceptions of sporting event impacts. People who live closer to the venues, to a certain extent, will feel discomfort due to the congestion and crowds caused by the event (Maksum et al. 2012). Our data suggested that gender had no association with all six variables indicating that both men and women tended to rate sports event image, motivation, satisfaction, stadium atmosphere, environment, and the perception of sports event impacts relatively the same.

### 4.4. Study Limitations and Further Study Directions

This study has some limitations. Data collection was conducted during the PON XX and this might contribute to the better results since respondents were still in a euphoric

atmosphere. Although several dimensions were examined, the image destination aspect was not investigated. Meanwhile, Papua has numerous international class destinations. With these limitations, further studies are recommended to (1) use a time series in the data collection before, during, and after the event, and (2) add other relevant variables, such as destination image, happiness in life, and other related constructs.

## 5. Conclusions

The implementation of the largest multi-event sport in Indonesia in 2021, the PON XX, was positively rated by the community in the aspects of sports event image, satisfaction, motivation, stadium atmosphere, environment, and the perception of the impact of sporting events. Based on the results, the highest motivation for people to watch the event was for entertainment. In addition, there was a strong relationship between PON XX and the six constructs. The direct involvement in the event was associated significantly with event image, satisfaction, motivation, stadium atmosphere, environment, and the perception of the impact of sporting events. The distance of the residence and the venues was associated with the perceived impact of the PON XX only and gender had no association with all six constructs.

This study provides practical implications that the implementation of sporting events, such as the PON needs to be held continuously by governments since the community tends to respond positively to such events and it could have positive impacts on economics. In addition, the existence of sports events might also be used to evaluate the development and progress of sports that have been carried out by each region in the country. In addition, our data suggest that public perception of the impact of sporting events is closely related to the variables of sports event image, motivation, satisfaction, stadium atmosphere, and stadium environment. Therefore, to achieve the high satisfaction of the community, the stadium atmosphere and stadium environment need to be the priority of the organizing committee.

**Author Contributions:** Conceptualization, K.K., T.S.G. and M.F.P.P.; Data Accuracy, M.F.P.P.; Formal analysis, K.K. and T.S.G.; Investigation, M.F.P.P.; Methodology, K.K. and T.S.G.; Supervision, T.S.G.; Validation, K.K. and T.S.G.; Writing—original draft preparation, K.K., T.S.G. and M.F.P.P. All authors have read and agreed to the published version of the manuscript.

**Funding:** This study received no external funding.

**Institutional Review Board Statement:** This study was conducted according to the guidelines of the Declaration of Helsinki. Ethical review and approval were also waived as this was an observational study that has no risk impact.

**Informed Consent Statement:** Informed consent was obtained from all respondents involved in the study.

**Data Availability Statement:** Data supporting presented results can be found by directly asking the correspondence.

**Acknowledgments:** The authors are grateful to all respondents who participated in this study.

**Conflicts of Interest:** The authors declare no conflict of interest.

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
