# Peer review of "Sports Event Image, Satisfaction, Motivation, Stadium Atmosphere, Environment, and Perception: A Study on the Biggest Multi-Sport Event in Indonesia during the Pandemic"

_socsci, doi:10.3390/socsci11060241_

Round 1
Reviewer 1 Report
Thank you for the opportunity to review. I believe that the manuscript is very well written, in accordance with the rules of writing scientific papers.
From my comments it is:
- the authors write that they used 6 instruments to evaluate each variable. Please describe it better. There is a lack of information on these six instruments. How to understand and describe this research procedure better.
- The part presenting the research results and part of the discussion is very long.
- The conclusions take up so little space in the text. What are some tips for event organizers, what are the indications for the future? Can the research results be used in a practical way?
- The part bibliography must be technically corrected.
Author Response
Dear Reviewer,
Please see the attachment.
Best regards,
Authors

Reviewer 2 Report
Thank you so much for giving me the opportunity to review this manuscript.
General Comments
The authors explored an interesting area of research related to the biggest multi-sport event in Indonesia, to understand the relationships of sports event image, satisfaction, motivation, stadium atmosphere, environment, and perception. However, the manuscript must be thoroughly revised to make it ready for publication.
Abstract
In the abstract, why National Sports Week was abbreviated a PON, rather than NSW? Could you please clarify? Please make a note for abbreviations if it was because of Indonesian name.
Introduction
P1. L. 34. What is PON XX?
In the introduction, in the first three paragraphs, the main discussion revolves around the selection of Papua, to conduct the National Sports Week. I would suggest discussing about the necessity of this study and expose the research gap.
Materials and Methods
Instrument
It is suggested, not to repeat the explanation of 5-point Likert scale ranging from 1-5, for the constructs, community’s motivation, stadium atmosphere, stadium environment and perception. Please weed out the sentences, that is with repetitive explanation.
Results
It is suggested to incorporate the detailed discussion of Table 6 and Table 7.
Discussion
Discussion must be based on the outcome of your results, rather than in a general manner.
It is suggested to include theoretical and practical implications of the study.
Discuss the limitations and areas of future research under separate heading.
Author Response

(The authors gave the same response as above.)

Round 2
Reviewer 2 Report
The authors substantially revised the manuscript and, maybe accepted for publication.
This manuscript is a resubmission of an earlier submission. The following is a list of the peer review reports and author responses from that submission.